# The Dynamic Changes of *Brassica napus* Seed Microbiota across the Entire Seed Life in the Field

**DOI:** 10.3390/plants13060912

**Published:** 2024-03-21

**Authors:** Yao Yao, Changxing Liu, Yu Zhang, Yang Lin, Tao Chen, Jiatao Xie, Haibin Chang, Yanping Fu, Jiasen Cheng, Bo Li, Xiao Yu, Xueliang Lyu, Yanbo Feng, Xuefeng Bian, Daohong Jiang

**Affiliations:** 1State Key Laboratory of Agricultural Microbiology, Huazhong Agricultural University, Wuhan 430070, China; yaoyao19950131@webmail.hzau.edu.cn (Y.Y.); 15827068697@163.com (C.L.); taochen@mail.hzau.edu.cn (T.C.); jiataoxie@mail.hzau.edu.cn (J.X.); boli@mail.hzau.edu.cn (B.L.); xiaoyu@mail.hzau.edu.cn (X.Y.); lvxueliang@mail.hzau.edu.cn (X.L.); fengyanbo0208@163.com (Y.F.); xuefengbian@webmail.hzau.edu.cn (X.B.); 2Hubei Key Laboratory of Plant Pathology, Huazhong Agricultural University, Wuhan 430070, China; zhangyu190919@163.com (Y.Z.); yanglin@mail.hzau.edu.cn (Y.L.); yanpingfu@mail.hzau.edu.cn (Y.F.); 3Hubei Hongshan Laboratory, Wuhan 430070, China; 4Huanggang Academy of Agricultural Science, Huanggang 438000, China; chang100362@163.com

**Keywords:** seed microbiota, seed development, *Brassica napus*, core microbiota, human pathogen, plant health, *Sphingomonas endophytica*

## Abstract

The seed microbiota is an important component given by nature to plants, protecting seeds from damage by other organisms and abiotic stress. However, little is known about the dynamic changes and potential functions of the seed microbiota during seed development. In this study, we investigated the composition and potential functions of the seed microbiota of rapeseed (*Brassica napus*). A total of 2496 amplicon sequence variants (ASVs) belonging to 504 genera in 25 phyla were identified, and the seed microbiota of all sampling stages were divided into three groups. The microbiota of flower buds, young pods, and seeds at 20 days after flowering (daf) formed the first group; that of seeds at 30 daf, 40 daf and 50 daf formed the second group; that of mature seeds and parental seeds were clustered into the third group. The functions of seed microbiota were identified by using PICRUSt2, and it was found that the substance metabolism of seed microbiota was correlated with those of the seeds. Finally, sixty-one core ASVs, including several potential human pathogens, were identified, and a member of the seed core microbiota, *Sphingomonas endophytica*, was isolated from seeds and found to promote seedling growth and enhance resistance against *Sclerotinia sclerotiorum*, a major pathogen in rapeseed. Our findings provide a novel perspective for understanding the composition and functions of microbiota during seed development and may enhance the efficiency of mining beneficial seed microbes.

## 1. Introduction

In 2020, the FAO raised questions about how to feed 10 billion people by 2050 without destroying natural resources. Furthermore, we also must face the destruction of crop production caused by continuous natural disasters, such as global warming, crop diseases, pests and biological invasions. Considering that developing new techniques to protect plant health and improving our environment are urgent tasks, researchers have suggested that natural resources, including the ‘phytomicrobiome’, are the most promising solution for achieving food security and ensuring environmental safety [1].

Microbes coexist and coevolve with plants; they are closely related to plant health and are considered holobionts [2]. Plants provide many niches, such as leaves, stems, roots and rhizosphere soil, for the growth and reproduction of a wide variety of microbes. Seeds, the beginning and the end of plant life, also carry numerous microbes [3,4]. Microbes in seeds affect food quality, seeding activity, seed health and sanitation [5,6]. Microbes isolated from seeds could promote plant growth [7,8,9,10,11] and enhance plant resistance [12,13,14,15]. Furthermore, seed microbes could be directly inoculated on seeds and protect plant (seedlings) health [4,16,17,18]. To the best of our knowledge, seed microbiota has been investigated in various important crops, such as rice, corn, barley, rapeseed, tomato and radish [9,19,20,21,22,23,24,25,26]. These findings have greatly inspired the continued investigation of seed microbiota, because seed microbes have enormous potential for protecting plant health.

The environment [27,28,29], plant genotype [30,31,32,33], development status of plants [34,35,36,37], different niches of plants [38,39,40], biotic and abiotic stress [41,42,43,44] and many other factors may affect the structure of plant microbiota. However, plant species have their own core microbiota, which not only exists stably but plays an important role in their environment [45,46,47,48]. By using a meta-analysis of 63 seeds of 50 plants, a stable core microbiota of bacterial and fungal taxa was found to be widely distributed, and was identified in most plant species [49]. This suggested that there was a long evolutionary relationship between plants and core microbiota.

The assembly of microbiota is likely to be finely controlled, and small differences in early colonization would result in significant differences in the structure of microbial communities, which is known as the priority effect [50,51]. Early colonizing microbes could preferentially occupy the ecological niche and resources and prevent other microbes from colonizing [52]. Seeds served as the endpoint for microbial assembly in the previous generation of plants and the starting point for the establishment of microbial communities in the next generation of plants, playing an important role in the assembly of the early plant microbiota and transmission of microbes between different generations [8,24,25,53,54,55,56,57]. Although vertical transmission of seed microbiota has been frequently observed, little is known about the composition of the core microbiota and the succession of microbes during seed development, and their function has rarely been investigated [58].

In this study, we used *Brassica napus*, a widely planted crop cultivated for edible plant oil, to probe the dynamic changes in microbiota during seed development, using 16S rRNA gene amplicon sequencing techniques. DNA of all samples was extracted from parental seeds, flower buds, young pods (10 days after flowering, daf), developing seeds (i.e., 20 daf, 30 daf, 40 daf and 50 daf) and mature seeds (seeds after harvesting and drying under environmental temperatures). We compared the structure of the seed microbiota at different developmental stages and identified the core microbiota. We also predicted the KEGG pathways of seed microbiota using PICRUSt2 [59] and compared them with the substance metabolism of rapeseed seeds. Then, we isolated and identified a core microbe from the seeds of rapeseed, *Sphingomonas endophytica*, and its role in the health of rapeseed seedlings was evaluated.

## 2. Results

### 2.1. Alpha and Beta Diversity among All Samples

Overall, we generated 4,159,158 reads of raw data, and the sequencing quality (Q30) of each sample was greater than 91% (Appendix A). After removing the sequences of mitochondria and chloroplast, a total of 3,054,394 counts were obtained. The dataset was rarefied to the lowest number of 17,546 reads and normalized (Total sum scaling, TSS) for analysis. The number of goods coverage of all samples was greater than 99% (Appendix A), indicating that our sequencing was sufficient to reveal the true diversity of all samples. Furthermore, to obtain the sequencing contamination in the background, such as that from DNA extraction kits and PCR reagents, an empty extraction was designed, which underwent the same process as the plant-related samples and served as negative control. The number of reads in the negative control samples ranged from 2 to 219 (Appendix A), and the variety of ASVs ranged from 0 to 11 (Appendix A), which was relatively lower than that of the seed-related samples. The lowest number of reads and the variety of ASVs among seed-related samples were 23,759 and 58, respectively. These results indicated that the process of operation was not heavily contaminated, and that the results were credible.

To evaluate the alpha diversity with respect to seed development and maturation, three commonly used diversity indices (i.e., observed features, evenness and Shannon index) were analyzed at the ASV level (Appendix A). The index of observed features represents the number of ASVs that could be observed in each sample. The group of flower buds had the lowest number of observed features compared to any other groups, and the highest number was found in mature seeds (Figure 1a). As for evenness, it reflects the evenness of the relative abundance of each ASV. The evenness of the mature seeds and parental seeds was significantly lower than that of the other six groups (Figure 1b). Finally, the Shannon index of each group was calculated; this takes into account both observed features and evenness. Higher values of the Shannon index indicate a higher alpha diversity of the microbial community. Analysis revealed that the Shannon index of the flower buds was significantly lower than that of any other group during seed development. With seed development, this gradually increased, and it reached its highest value in 50 daf seeds (Figure 1c). When seeds were harvested and dried naturally, the Shannon index significantly decreased.

Principal coordinates analysis (PCoA) and UPMGA clustering were carried out based on the Bray–Curtis distance (Figure 1d,e). The results showed that the *p* value between any two groups was less than 0.05, except for the groups of parental seeds and mature seeds (*p* value = 0.56) (Appendix A). All samples were mainly clustered into three groups (Figure 1e). The flower buds, young pods and 20 daf seeds were clustered into one group. Seeds that had developed for 30, 40 and 50 daf were clustered into another group. With seed development, a unique seed microbiota formed gradually. The third group comprised the samples of mature seeds and parental seeds. After drying and dehydration, the structure of the seed microbiota was similar between two generations.

### 2.2. Composition and Differences in Seed Microbiota at the Phylum and Genus Levels

The bacterial community consisted of 25 phyla. Each group was mainly composed of Gammaproteobacteria, followed by Actinobacteria, Firmicutes, Alphaproteobacteria, Batceroidetes and Fusobacteriota (Figure 2a, Appendix A). Difference analysis was conducted, and a total of 8 differential taxa were obtained (Figure 2b; Appendix A). The groups of flower buds, young pods, and 20 daf seeds were enriched in Bacteroidetes and Fusobacteriota; seeds at 40 daf and 50 daf were enriched in Alphaproteobacteria; mature seeds and parental seeds were enriched in Gammaproteobacteria and unclassified Proteobacteria.

A total of 504 genera were analyzed among all samples. The top 20 genera in relative abundance are displayed (Figure 2c), and the top 10 genera across each group were also determined (Figure 2e–l; Appendix A). For the three groups of early stages (flower buds, young pods and 20 daf seeds), the genera *Actinomyces*, *Streptococcus*, *Brevundimonas*, *Haemophilus*, *Prevotella*, *Alloprevotella* and *Neisseria* were all present in the top 10 (Figure 2e–g). At the sampling time points of 30, 40 and 50 daf, the genera *Staphylococcus*, *Corynebacterium*, *Actinomyces* and *Streptococcus* were all present in the top 10 (Figure 2h–j). For the two groups of mature seeds and parental seeds, genera in the top 10 were *Ralstonia*, *Bosea*, *Pseudomonas*, unclassified genera in Comamonadaceae and Caulobacteraceae, *Actinomyces*, *Ochrobactrum* and *Delftia* (Figure 2k,l). Difference analysis was conducted at the genus level, and a total of 25 differential genera were obtained (Figure 2d; Appendix A). Compared to the microbiota at the early stages (flower buds, young pods and 20 daf seeds), the genera *Bacillus*, *Sphingomonas* and so on were significantly enriched in the microbiota at the late stages (seeds 30, 40 and 50 daf). For the two groups of parental seeds and mature seeds, the genera *Ralstonia* and *Delftia* were significantly enriched.

### 2.3. Functions of Seed Microbiota during Seed Development of Rapeseed

Microbial functions on developing seeds were predicted by PICRUSt2. The functions of each developmental stage were mainly composed of amino acid metabolism at the second level (Figure 3a; Appendix A). The differential pathways are displayed, and the heatmap clustering confirmed that early developmental stages (flower buds, young pods and 20 daf seeds) were clustered into one group; late developmental stages (seeds at 30, 40 and 50 daf) were clustered into another group (Figure 3b), which was consistent with the above results (Figure 1e and Figure 2b,d).

At the third level of KEGG pathways, the top 40 in relative abundance are displayed (Figure 3c; Appendix A), and the differential functions of developmental seeds (seeds 20, 30, 40 and 50 daf) were also analyzed. Considering that lipid metabolism, carbohydrate metabolism, amino acid metabolism and biosynthesis of other secondary metabolites play important roles in the development of seeds, we focused on the above four differential pathways (Figure 3d,e). According to previous research, oil synthesis of rapeseed seeds began at approximately 20 daf and peaked at 40 daf [60], with a decreasing trend in the later stages of seed maturation [61]. In this work, fatty acid biosynthesis was enriched at 20 daf; biosynthesis of unsaturated fatty acids, alpha-linolenic acid metabolism and fatty acid metabolism was enriched at 40 daf; fatty acid degradation was enriched at 50 daf. It was speculated that microbial oil synthesis of seed microbiota began at 20 daf, accumulated at 40 daf and decreased at 50 daf, which was consistent with the trend of oil synthesis of rapeseed seeds. Starch metabolism was also a potential factor leading to high oil-content. During the period of 20–45 daf, high oil-content rapeseed seeds have higher levels of starch and sucrose compared to low oil-content seeds [62,63]. In this study, the 20 daf seed microbiota was significantly enriched in starch and sucrose metabolism, suggesting that the seed microbes at 20 daf were equally important for oil content.

As for substances other than oil, a previous study has analyzed the transcriptome of rapeseed seeds at 24 and 33 daf, and pyruvate metabolism was more active, indicating a greater tendency toward pyruvate utilization during this period [64]. In this study, pyruvate metabolism was also significantly enriched at 30 daf. The seed coat color is one of the important characteristics of rapeseed seeds. The transcriptomes between yellow-coat seeds and black-coat seeds were compared, and the differentially expressed genes (DEGs) were involved in phenylpropanoid biosynthesis and phenylalanine metabolism [65]. In this study, phenylpropanoid biosynthesis and phenylalanine metabolism were enriched at 50 daf. During the period from 40 to 50 days, the color of the rapeseed seed coat changed from cyan to brown. Screening the differential microbes related to these pathways may be valuable for rapeseed seed coat color. Glucosinolate is one of the important secondary metabolites of rapeseed [66] in plant resistance [67]. A study showed that the glucosinolate content in rapeseed seeds at 25 daf increased by 63% compared with that at 15 daf [68]. In this study, glucosinolate biosynthesis was significantly enriched at 20 daf. The above results suggested that the differential functions of seed microbiota were related to those of the seeds.

In this study, flavonoid-related functions also existed, such as flavone and flavonol biosynthesis, flavonoid biosynthesis, and isoflavonoid biosynthesis. However, there were no significant differences among different groups for the above pathways, and the relative abundances were less than 0.01%. It was speculated that the contribution of seed microbiota to flavonoids was limited.

### 2.4. Identification of Seed Core Microbiota from Seed Sowing to Maturation

Out of 2496 ASVs, a total of 61 ASVs were present in each group, this set was identified as the core microbiota (Figure 4a; Appendix A). Furthermore, the relative abundance of each core ASV was presented in the Appendix A. These 61 core ASVs belonged to five phyla and thirty-seven genera. The phylum and genus with the highest microbial diversity were Proteobacteria and *Streptococcus* (Figure 4b,c). The means of the total relative abundance of the core microbiota were determined and compared among all groups, and it accounted for the major component in parental seeds, reaching 77.08%. In other groups, it was 75.06% in flower buds, 68.47% in mature seeds, 65.69% in young pods, 60.65% in 20 daf seeds and 49.45% in 30 daf seeds (Figure 4d). For the 40 daf seeds and 50 daf seeds, it was relatively lower, only 29.97% and 34.62%, respectively, and the unique ASVs became the main component. Statistical analysis was also conducted, and a total of 42 differential core ASVs were obtained (Figure 4e). Unexpectedly, *Neisseria*1 and *Neisseria*2 were enriched in the early three stages. The highest mean of total abundance among all groups was in flower buds (8.22%), followed by 20 daf seeds (6.37%) and young pods (3.99%), and the lowest was in mature seeds (0.31%) (Appendix A). Notably, Neisseriaceae is one of the common families in pollinator microbiomes [69,70]. The above findings apply equally to *Prevotella*1 and *Prevotella*2 [69,71], which were also enriched in flower buds (4.62%), followed by young pods (1.62%), 20 daf seeds (1.59%), and found to be decreased in mature seeds (0.02%).

### 2.5. The Differences in Seed Microbiota between Developmental and Maturity Stages

Considering that microbes may influence seed quality, we focused on the core microbiota in the process of seed formation. The number of common ASVs increased from 61 to 89 after removing the groups of mature seeds and parental seeds (Figure 5a). For the total relative abundance of 28 missing core ASVs, this value ranged from the lowest (3.22) to the highest (6.58), which was relatively uniform (Figure 5b,c; Appendix A). In this study, the samples of negative control were used to evaluate contamination (Appendix A). Compared to the core sixty-one ASVs, a total of seven common ASVs were counted (Figure 5d). As for the groups of mature seeds, parental seeds and developing seeds, 190 common ASVs existed in these three groups (Figure 5e). Although the number of common ASVs was lower than that of unique ASVs, it demonstrated a high relative abundance. Across seed microbiota of two generations, there were 353 common ASVs in mature seeds and parental seeds. These were dominant in relative abundance, accounting for 93.75% and 89.06% in parental seeds and mature seeds, respectively, suggesting that the assembly of the seed microbiota was not random. Furthermore, we obtained the differential ASVs between 50 daf seeds and mature seeds. The heatmap showed that all samples were mainly clustered into two groups (Figure 5f). As to microbial diversity, a total of 11 *Pseudomonas*-related ASVs were enriched in mature seeds, followed by *Ralstonia*, with 10 kinds of ASVs (Figure 5g). The above results suggest that the dominant taxa in mature seeds may be enriched during the process of maturation.

### 2.6. Exploring Microbial Resources of Rapeseed Seeds

In the sequencing data, the core ASV *Sphingomonas*1 had shown low relative abundance in the early stage of seed formation, only 0.04% and 0.13% in 20 daf and 30 daf seeds, respectively. With seed development, its abundance increased and remained stable, but the highest relative abundance was only 0.46%, in 40 daf seeds (Appendix A). The bacterial strain was isolated from rapeseed seeds and presented a yellow, circular and smooth-edge morphology (Figure 6a). The 16S rRNA gene sequence of this strain was matched with that of core ASV *Sphingomonas*1, with a resulting similarity of 98.41% (372/378) (Appendix A). The above value was more than 97%, which was the common threshold in the analysis of microbiota. Furthermore, it was not one of the contaminating sequences in the group from the negative control. Phylogenetic analysis supported the finding that this strain is a member of *Sp. endophytica* (Figure 6b). Rapeseed seeds were inoculated with *Sp. endophytica* by seed biopriming, and the results showed that inoculation had a great effect on promoting the fresh weight of aerial parts of seedlings but did not affect the root length under Hoagland medium (Figure 6c,d). Under soil conditions, *Sp. endophytica* had a good effect on promoting the fresh weight of aerial parts and root length (Figure 6e,f). Then, we explored the resistance of rapeseed seedlings to *Sc. sclerotiorum*. The results indicated that *Sp. endophytica* could not only promote the growth of rapeseed seedlings but also enhance resistance to *Sc. sclerotiorum* (Figure 6g,h).

## 3. Discussion

Seed microbes are particularly noteworthy because they can be transmitted vertically and connect with different generations. Our study determined the dynamic changes and core microbiota of rapeseed seeds, from seed sowing to harvesting, and emphasized the consistency between seed microbial functions and the metabolism of seeds. Our findings provide a comprehensive understanding of the compositions and functions of seed microbiota and provide a reference for mining and applying beneficial microbes at different stages of seed development.

### 3.1. Dynamic Changes in the Composition of Seed Microbiota of Rapeseed, from Sowing to Harvesting

The seed microbiota is influenced by various factors, such as low temperature, rainfall, light, pollinator visitation, pathogen infection and the stage of seed development, which could lead to changes in seed microbiota [72]. Furthermore, the seed microbiota can be acquired from the environment either prior to or after the maturation of the seeds (horizontal transmission), or the offspring can take up the microbiota from the parents (vertical transmission) [73]. As to the structure of seed microbiota, that of the 20 daf seeds was more similar to those of flower buds and young pods, suggesting that early seed microbiota was derived from plants to a great extent, and that it may be influenced by vertical transmission from plant to seed and horizontal transmission by pollinators through the flower. The seed microbiota at 30 daf was relatively independent, while those at 40 and 50 daf were more similar (Figure 1d,e); the specific structure of the seed microbiota was formed gradually. Furthermore, the genera *Bacillus* and *Sphingomonas* were enriched in 40 daf and 50 daf, respectively (Figure 2d). Both of these genera had significant effects on plant resistance [74,75,76,77,78], indicating the possibility of resistance of the genera to environmental changes and biotic stresses during this period. During the process of seed maturation, seed microbiota also changes due to a series of influences such as water loss and high osmotic pressure [58]. In this study, when rapeseed seeds were detached from pods and lost water naturally, the microbiota of mature seeds changed compared to that of 50 daf seeds, but it was similar to that of the parental seeds (Figure 1d,e), suggesting that some microbes not only exist in seeds, but maintain a stable abundance, even in different generations.

### 3.2. The Function of Microbiota Is Consistent with the Accumulation of Seed Metabolites during Seed Development

In this study, we used PICRUSt2 to predict the functions of seed microbiota; this prediction program was based on microbial genomes that were already in existence and is mostly used for human microbiota. This program may have some limits for predicting functions of plant microbiota, and we identified pathways such as “endocrine system” and “primary bile acid biosynthesis”. Obviously, these pathways only exist in humans and animals, not in plants. With some exceptions, PICRUSt2 could be used for the prediction of plant microbiota. Plants exude different compounds depending on developmental stage, and in return, these exudates aid in assembling specific microbiota for plants [37]. The recruitment of microbes remains one of the hotspots in the field of plant–microbe interactions [79]. Chaparro et al. proposed the idea that plants can select a subset of microbes at different developmental stages, presumably for specific functions [37]. Furthermore, these microbes may in turn promote the synthesis of corresponding substances. Here, we successfully used PICRUSt2 to predict many pathways related to rapeseed seed metabolism, such as fatty acid biosynthesis, biosynthesis of unsaturated fatty acids, alpha-linolenic acid metabolism and fatty acid metabolism, starch metabolism, pyruvate metabolism and biosynthesis of glucosinolates and phenylpropanoid. The high content of oil is a characteristic of rapeseed seeds, while starch and pyruvate are important for fatty acid synthesis [62,63,80], and the secondary metabolites like glucosinolates and phenylpropanoid are important substances for plants [65,68].

### 3.3. Core Microbiota from Sowing to Maturation of Rapeseed Seeds

We identified a core microbiota composed of 61 ASVs from a total of 2496 ASVs (Figure 4a). These 61 ASVs were microbes that coexisted and coevolved with rapeseed seeds, and they may have specific functions. For example, *Sphingomonas* spp. [15,74,75]; members of the family Comamonadaceae [81,82]; *Brevundimonas* spp. [83,84]; *Allorhizobium*-*Neorhizobium*-*Pararhizobium*-*Rhizobium* [85,86,87]; *Stenotrophomonas* spp. [41,88,89]; *Bacillus* spp. [77,78,90,91] and such all have been reported to be beneficial to plant health. Some species of *Neisseria* and *Prevotella* are pathogenic bacteria for both humans and animals [71,92], but are members of the seed core microbiota. They accounted for high relative abundance, especially in the flower stage, and this abundance decreased at the remaining sampling points, with the lowest abundance found in mature seeds. Furthermore, Neisseriaceae and Prevotellaceae are common families among pollinating insects [69,70], and pollinating insects could transmit bacteria into seeds [93]. All plants were grown naturally in the field, and strict surface sterilization was carried out to rule out human contamination. The above results suggested that bacteria may infect humans and animals and cause diseases by pollinating insects or plant-seeds indirectly. The presence of potential human and animal pathogens on plants also indicates that the health of humans on Earth is related to the health of animals, plants and the environment. In fact, this is the concept of “One Health” [94]. Microorganisms play a crucial role and are more interconnected than previously thought. In another study, it was determined that *Prevotella* spp. are highly capable of degrading cellulose in the human gut [95]. The genus *Prevotella* in seeds may have the same role in breaking down cellulose. Whether *Neisseria* and *Prevotella* on rapeseed are pathogens for humans or animals, as well as determining their impact on the plants themselves, are topics which require further investigation.

Like other researchers, we also found a high abundance of *Ralstonia* in rapeseed seeds [20,21]. The genus *Ralstonia* has various members, including plant pathogens [96,97], biocontrol agents [98] and pathogens of humans and animals [99,100]. Interestingly, some *Ralstonia* could synthesize polyhydroxybutyrate (PHB) [101,102], which is one of the components of rapeseed seeds [103], and can be purified from seed meal [104]. We attempted to isolate *Ralstonia* from seeds, but were unsuccessful. Fortunately, we isolated *Ralstonia* from rapeseed root, with a 100% sequence identity to core ASV *Ralstonia*, and it was identified as *R. pickettii* through Blastn analysis (Appendix A). *R. pickettii* is a human pathogen [99], particularly in immunocompromised patients [105], and it is also a biocontrol agent for controlling the bacterial wilt of tomato [98]. The enrichment of *R. pickettii* in the gut is related to the decrease of unsaturated fatty acid levels [106]. In plants, unsaturated fatty acid negatively affects the lifespan of seeds due to peroxidation [107]. *R. pickettii* may also be responsible for the reduction in the level of unsaturated fatty acid in rapeseed seeds, and this needs to be investigated. To sum up, some microbes are both potential pathogens for humans and stable in rapeseed seeds. It has been discovered that the phyllosphere serves as a niche for human pathogens [108], and a human pathogen could colonize the entire plant for up to 21 days after inoculation [109]. The connection between plant and human microbiota deserves more attention in the context of “One Health” [94].

### 3.4. Unique and Missing ASVs during Seed Development and Maturation

To study the vertically transmitted ASVs, 353 ASVs common to parental seeds and mature seeds were identified, and these ASVs maintained a high abundance, even in different generations (Figure 5c). Interestingly, 163 ASVs were present in two generations, but not in the developing seeds; some microbes could enter into seeds, but had not been transmitted vertically by parental seeds. With seed maturation, 28 ASVs were present in developmental stages, but not in dry seeds (Figure 5b,c). Missing core ASVs lived in seeds for a period of time, but they could not be passed on to mature seeds. It was hypothesized that these microbes are relevant for the previous generation but may not be important for the next generation of seedlings. Moreover, there were seven ASVs common to the core microbiota and the contaminating ASVs (Figure 5d). These seven ASVs may be environmental microbes or seed core microbiota that were contaminated by aerosols. Even if these seven ASVs were excluded, fifty-four core ASVs could accurately be identified as seed core microbiota.

### 3.5. Providing a Basis for the Application Period of Microbes Sufficient to Establish Healthy Seed Microbiota

A study on endophytes in rice seeds showed that *Sp. melonis* accumulated in disease-resistant rice seeds and was transmitted across generations [15]. In another study, researchers inoculated specific endophytic microbes at the flowering stage and found that microbes can be vertically transmitted to the next generation [110]. These results showed that it was possible to apply beneficial microbes during the flowering period and achieve microbial breeding [111]. As shown in this study, microbes may play important roles in the metabolism of seed materials (Figure 3). Therefore, applying microbes at specific stages may enhance the content levels of required components in seeds. For example, to obtain seeds with high oil-content, we might pay attention to the enriched microbes in 20 daf seeds, because they are likely involved in fatty acid biosynthesis (Figure 3d).

### 3.6. Application of Core Microbes

Utilizing plant microbes, especially endophytic and rhizosphere microbes, could be a more efficient tactic towards improving plant health [112]. Furthermore, core microbes could also be utilized to effectively promote plant productivity [113]. In this study, we isolated *Sp. endophytica* from rapeseed seeds, one of the core endophytic microbes. The genus *Sphingomonas* was also one of the seed core bacterial members among 50 kinds of plant species [49]. When inoculated on rapeseed seeds, it could improve seedling growth and resistance to *Sc. Sclerotiorum* (Figure 6); this suggests that it is crucial to determine the functions of stable seed microbes with low relative abundance as to plant health. Furthermore, *Sp. endophytica* also exists in flower buds (Figure 4c), indicating that it could be applied during the flowering season. Additionally, we also need to give attention to the remaining 60 core ASVs in rapeseed seeds. Further investigation of core microbes should be carried out to understand their functions and develop new bioinoculants for improving plant health.

## 4. Methods

### 4.1. Plant Growth and Sample Collection

Seeds of rapeseed cultivar Huashuang 4, a commercial double-low cultivar widely grown locally, were divided into two parts: one part was used to extract DNA, the other was sown in a rapeseed field located in Huanggang city, Hubei Province, PR China (114.922567, 30.571612), in September 2020. The field had undergone rapeseed–rice rotation for more than ten years and was conventionally managed without the use of any fungicides. In March 2021, during the flowering season, for each individual plant, two adjacent flower buds that were about to open were marked with color thread and regarded as one biological replicate. Furthermore, some flowers were sampled to extract DNA, and 200 nonadjacent plants in total were randomly chosen for the labeling of flower buds. Ten days after flowering, young pods that appeared on the marked flowers were randomly collected. Because of the small size of the seeds, it was very difficult to pick seeds from the pods at 10 daf. After that, pods that grew for 20, 30, 40 and 50 daf were randomly collected, and seeds were picked from these pods. Pods collected from 6–8 plants were obtained at each sampling time and each plant was regarded as one independent biological replicate. All picked flower buds and pods were chilled in an ice box and brought back to the laboratory. The rapeseed seeds were also harvested from the sampling field in a 2 m × 5 m plot at 50 daf and were regarded as one biological replicate. Seeds from six plots of equal size were collected. All seeds underwent air drying naturally for one month; such seeds were regarded as mature seeds.

Alcohol-burned blades and tweezers were used to peel rapeseed seeds from pods (seeds at 20 daf, 30 daf, 40 daf and 50 daf). For all materials, the surfaces of all samples were disinfected by soaking in 75% alcohol for 30 s, in 3.5% bleach for 1 min, and then rinsing with sterilized water three times, each time for 1 min. Samples were frozen using liquid nitrogen and stored at −80 °C. In addition, the effect of surface disinfection was inspected. One hundred microliters of sterile water was placed on R2A medium (Hopebio, Qingdao, China), and if no bacteria grew after 5 days, it was determined that the surface was disinfected and free from contamination.

### 4.2. DNA Extraction, PCR Amplification and Sequencing

Frozen samples were physically disrupted with a sterile mortar and pestle using liquid nitrogen. FastDNA^®^ SPIN for soil kit (MP Biomedicals, Irvine, CA, USA) was used to extract total DNA according to the manufacturer’s instructions. Furthermore, a negative control was designed; this was the sample that did not contain plant tissues but did undergo the same process of seed-related samples. Library generation was performed by two-step PCR, which was modified according to the original research [114]. DNA concentrations were measured by Nanodrop 2000 (Thermo Fisher Scientific, Waltham, MA, USA) and diluted to 20 ng/µL as templates. One microliter of DNA was used to construct the sequencing library. The V5-V7 region of the 16S rRNA gene was amplified with the PCR primers 799F (5′-AACMGGATTAGATACCCKG-3′) [115]-1193R (5′-ACGTCATCCCCACCTTCC-3′) [116]. The 16S rRNA gene amplification program was performed according to the research described in [30]: 98 °C for 3 min; the region was then amplified by 30 cycles of 98 °C for 20 s, 55 °C for 20 s and 72 °C for 1 min, followed by 5 min at 72 °C. The 50 µL PCR system contained 10 µL of 5 × buffer, 5 µL of dNTPs, 1 µL of DNA, 0.5 U of TransStart FastPfu DNA Polymerase (TransGen Biotech, Beijing, China) and each forward and reverse primer at 10 nM, while the remaining volume was supplemented with water to 50 µL. The amplification of first-step PCR was subjected to gel electrophoresis on 1.0% agarose. If no bands were visible in the negative control, the 500 bp target band was purified by an E.Z.N.A. Gel Extraction Kit (Omega Biotek, Norcross, GA, USA). The second-step PCR used the same procedure as the first-step PCR, but with only 8 cycles. Primer pairs with specific labels were used to construct a complete library structure; the primer sequences are shown in (Appendix A). If no amplification was visible from the negative control, magnetic beads (Yeasen Biotechnology, Shanghai, China) were used to purify the product. The purified products were diluted to 20 ng/µL and mixed with 2 µL for each sample. Finally, the sequencing library was sent to the Company (Genewiz, Suzhou, China) and subjected to paired-end 250 bp (PE 250) sequencing on the NovaSeq 6000 platform.

### 4.3. Bioinformatics Analysis of 16S rRNA Gene Profiling

Sequencing data were analyzed by QIIME2-2021.2 [117]. Based on DADA2 of QIIME2 [118], priming sequences were trimmed out according to the number of bases, and the forward and reverse sequences were spliced to obtain the composition of the feature table for analysis. The 16S rRNA gene sequences were annotated by comparing version 138 of the SILVA database [119,120]. Before analyses, sequences matching to ‘chloroplast’ and ‘mitochondria’ were removed by the method of taxonomy-based filtering implemented in QIIME2-2021.2. For the groups of developing seeds (flower buds, young pods, 20 daf seeds, 30 daf seeds, 40 daf seeds and 50 daf seeds), data optimization was carried out and the sequences that were present in at least 2 samples and occurred with a total frequency greater than 5 were retained. For the two groups of mature seeds and parental seeds, due to the high relative abundance of ASV *Ralstonia*, the data were not optimized in order to avoid overlooking the ASVs with low relative abundance. After this step of analysis, we deleted the sequence that was only annotated to the domain of “bacteria” among all samples. All of the feature table was normalized at 17,546 counts and the subsequent analysis was conducted based on the normalized table. The alpha diversity index was determined by QIIME2 (observed features, evenness, Shannon index and good coverage) at the ASV level. Our sequence processing in QIIME2-2021.2 followed the official tutorials of QIIME2 (https://docs.qiime2.org/2021.2/tutorials/ (accessed on 14 September 2021)). One-way ANOVA was used to calculate whether there were significant differences among different groups, and *p* value was taken to be the statistical indicator. Beta-diversity was calculated based on Bray–Curtis distance. Permutational multivariate analysis of variance (PERMANOVA, Adonis) was used to evaluate whether there were differences in any two groups in the vegan package in R-3.6.1. The composition of taxa at the phylum and genus levels was generated based on QIIME2. Alpha diversity boxplots, principal coordinate analysis (PCoA) and UPGMA trees were drawn by R-3.6.1 in the vegan and ggplot2 packages. Detailed parameters of analysis in R can be found in (Appendix A). Differential taxa were analyzed by LEfSe [121] and shown through a heatmap drawn by TB tools [122]. The functions of seed microbiota were obtained by PICRUSt2 [59] through an online website (https://www.bioincloud.tech/ (accessed on 8 June 2023)). Histograms of composition and functions were drawn by an online website (https://www.chiplot.online/ (accessed on 8 June 2023)). The common and unique ASVs were rendered by Venn diagram through an online website (http://bioinformatics.psb.ugent.be/webtools/Venn/ (accessed on 12 April 2023)). Core microbiota was obtained among all groups at the ASV level by an online website (http://www.cloud.biomicroclass.com/CloudPlatform (accessed on 3 April 2023)). We used a flower diagram to obtain the overlapping ASVs which exhibited more than 1 read in each group and identified as core microbiota.

### 4.4. Isolation and Identification of Seed Bacteria

Isolation of seed bacteria followed a method described by [123], with minor modifications. Mature seeds, the endpoint of the previous generation and the starting point of seedlings, were collected to isolate bacteria, and epiphytic seed bacteria were removed by surface sterilization (see above, describing plant growth and sample collection). A total of 0.2 g of seeds was ground in 1 mL of 0.01 M PBS buffer (Servicebio, Wuhan, China), and serial dilutions (10^−1^, 10^−2^, 10^−3^) were made. One hundred microliters of homogenate were plated on R2A medium. After 5 days, colonies of bacteria were transferred to new R2A medium.

DNA extraction of bacteria was performed according to a previous study [124]. The 16S rRNA gene sequence of bacteria was amplified using primer pairs 27F (5′-AGAGTTTGATCCTGGCTCAG-3′) and 1492R (5′-GGTTACCTTGTTACGACTT-3′). The PCR program was the same as that of the one-step PCR described in this article. PCR products were sequenced at the Company (Tianyihuiyuan, Wuhan, China). The sequences were BLASTn searched in the NCBI database, and the 16S rRNA gene sequences were all downloaded from NCBI (Appendix A). Phylogenetic analysis was carried out to evaluate the evolutionary relationships, using the method of neighbor-joining. Phylogenetic analyses were conducted in MEGA11.

To obtain the colony morphology of the strain *Sp. endophytica*, the strain was activated in R2A medium for two successive generations to ensure a lack of contamination and grown for 5 days at 25 °C. Then, colony morphology was observed by stereomicroscopy Leica M205FA (LeicaMicrosystems, Weztlar, Germany).

### 4.5. Plant Treatments and Assays

Strain *Sp. endophytica* was activated in R2A medium for two successive generations. The R2A medium was washed with 0.01 M PBS solution to obtain the bacterial liquid, and the OD_600nm_ was diluted to 0.4 for inoculation in rapeseed seeds. Seed inoculation of bacteria followed a method described by [18], with minor modifications. Rapeseed seeds were surface-sterilized in 75% alcohol for 1 min, 3.5% bleach for 3 min and finally rinsed with sterile water 3 times for 1 min each time. The inoculated seeds were soaked in the bacterial suspension for 4 h, and the non-inoculated seeds were treated with 0.01 M PBS solution. The seeds were transferred to Hoagland medium (Hopebio, Qingdao, China) for seed germination. One day later, seedlings with the same germination conditions were selected and transferred to 50 mL Hoagland medium or soil conditions (Peilei Substrate Technology, Zhenjiang, China). Seedlings grew in a greenhouse with alternating light and dark for 12 h/12 h at 20 °C for 2 weeks.

### 4.6. Plant Growth Promotion Assay

Plant growth promotion assays were performed following a method described by [18]. After 2 weeks, the fresh weight of aerial parts and root lengths of seedlings were calculated, respectively. The experiment was repeated three times, with 6–8 seedlings per biological replicate. Data analysis was based on one-way ANOVA (*p* value < 0.05) and conducted in Excel 2019 software. The symbol “*” indicates *p* value < 0.05, the symbol “**” indicates *p* value < 0.01, the symbol “***” indicates *p* value < 0.001 and the symbol “NS” indicates *p* value > 0.05.

### 4.7. Antagonistic Activities of Bacteria against Sc. sclerotiorum

The assay of antagonistic activities of bacteria against *Sc. sclerotiorum* followed a method described by [18]. Seedlings growing in soil were inoculated with *Sc. sclerotiorum* strain EP-1PNA367, which was cultured continuously in Potato Dextrose Agar (PDA) medium for 2 generations. Mycelial agar (2 mm diameter) was inoculated onto leaves that were taken from the margin of growing hyphae. All rapeseed seedlings were placed in a plastic basin and sealed with plastic wrap for moisturizing. Necrosis areas were counted after 24 h to evaluate the resistance of seedlings to *Sc. sclerotiorum*. The experiment was repeated three times with 6–8 seedlings per biological replicate. Data analysis was the same as in the above assay (the plant growth promotion assay).

## Figures and Tables

**Figure 1 plants-13-00912-f001:**
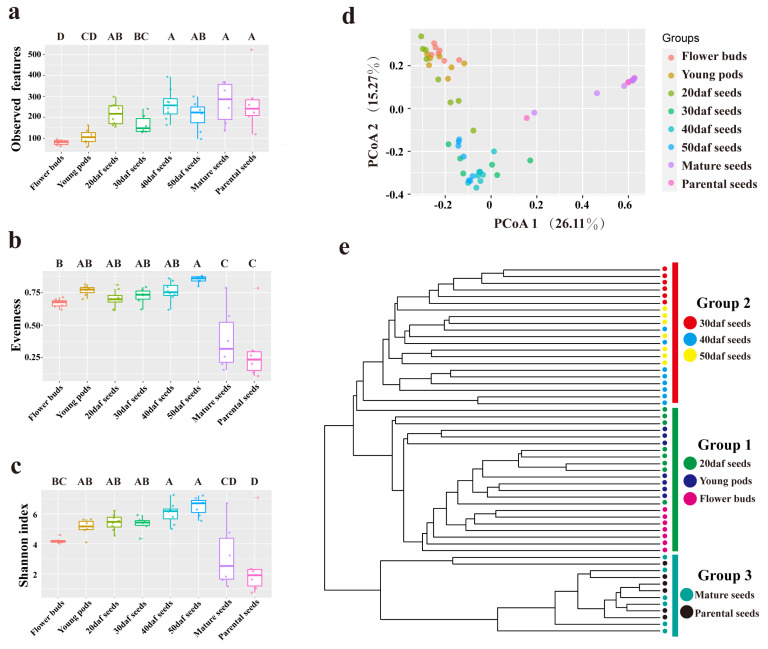
Alpha and beta diversity of seed microbiota. (**a**) Microbial diversity, (**b**) evenness and (**c**) Shannon index among all groups. Statistical significance was determined by one-way ANOVA (Duncan’s test). Letters (ABCD) represent the significant differences at the 95% confidence interval (*p* < 0.05). The horizontal line within each box represents the value for the median, the tops of boxes represent 75th percentile values, and the bottoms of boxes represent 25th percentile values. The upper and lower error bars encompass data varying no more than 1.5× interquartile range from the upper edge and lower edge of the box, respectively. (**d**) Principal coordinates analysis (PCoA) and a (**e**) UPMGA clustering tree based on Bray–Curtis distance show that the seed microbiota was mainly distributed in three parts. Flower buds, endophytic microbiota of flowers; Young pods, endophytic microbiota of pods grown for 10 days after flowering; Seeds 20, 30, 40 and 50 daf, endophytic microbiota of seeds grown for 20, 30, 40 and 50 days after flowering; Mature seeds, endophytic microbiota of seeds that underwent air drying naturally; Parental seeds, endophytic microbiota of seeds to be sown.

**Figure 2 plants-13-00912-f002:**
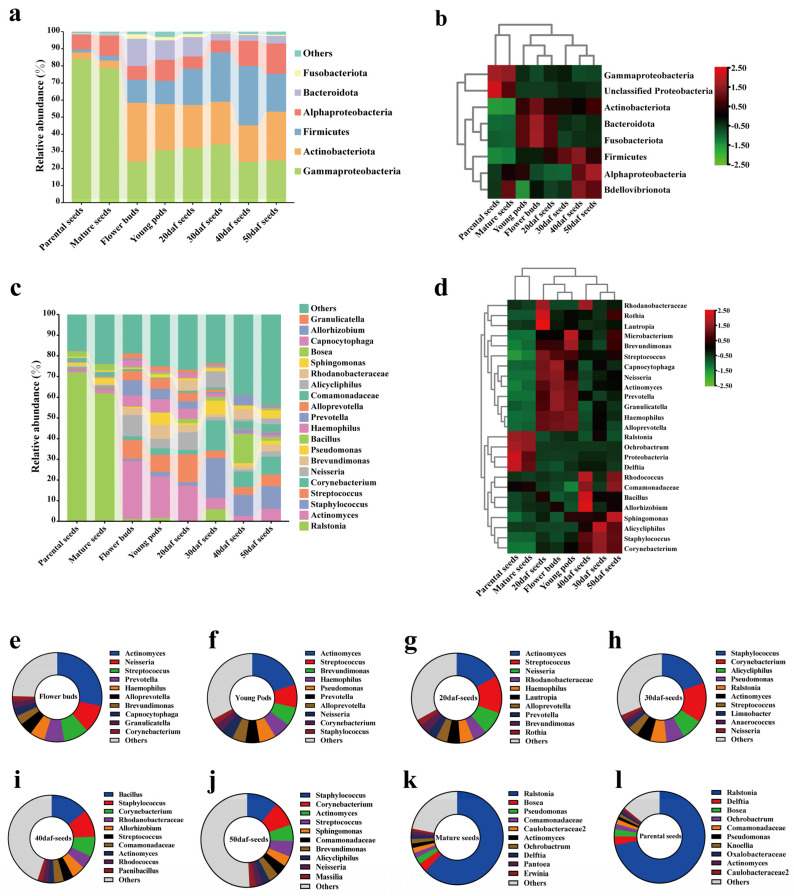
Composition and differential taxa of seed microbiota at the phylum and genus levels at different sampling stages. (**a**,**c**) Histograms of relative abundances in parental seeds, flower buds, young pods, developing seeds and mature seeds at the phylum and genus levels. Phyla with relative abundances greater than 1% and the top 20 genera in relative abundance, respectively, are shown in the diagram. (**b**,**d**) Heatmap showing differential taxa at the phylum and genus levels across all groups, respectively; analysis of differences was based on LEfSe. Thresholds were determined at the phylum level (LDA = 2, *p* < 0.05) and at the genus level (LDA = 4, *p* < 0.05). (**e**–**l**) Schematic diagram of the top 10 taxa of microbiota in flower buds, young pods, 20 daf seeds, 30 daf seeds, 40 daf seeds, 50 daf seeds, mature seeds and parental seeds at the genus level. See Figure 1 for the details of flower buds, young pods, 20, 30, 40 and 50 daf seeds, mature seeds and parental seeds.

**Figure 3 plants-13-00912-f003:**
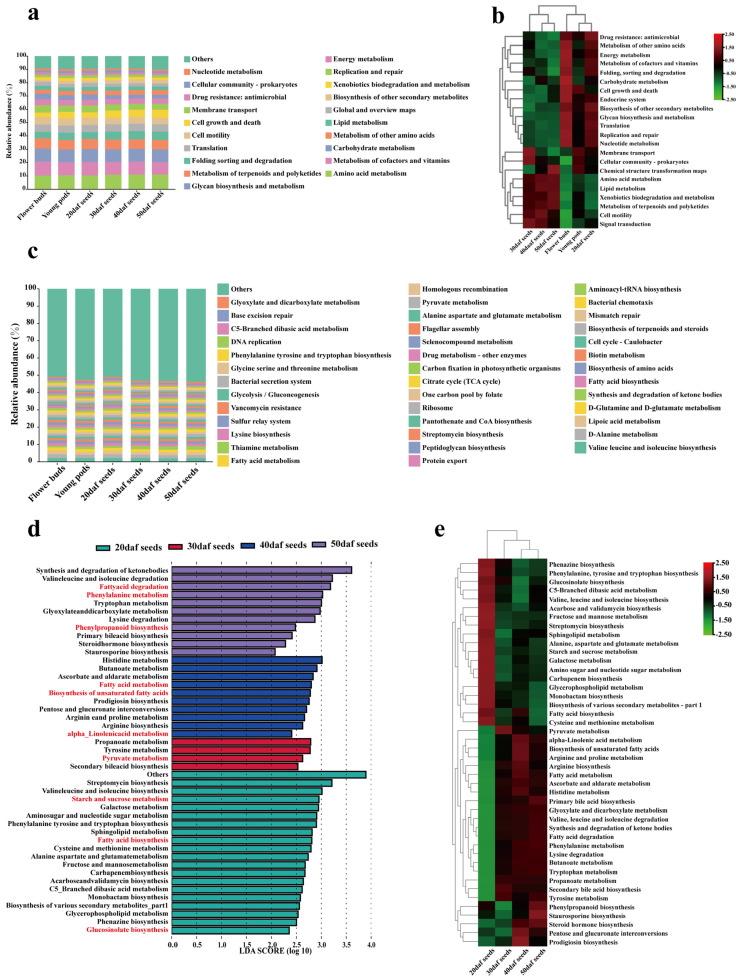
Functions of seed microbiota during seed development based on PICRUSt2. (**a**) The prediction of the top 20 KEGG pathways in each group at the second level. (**b**) Heatmap showing differential functions with relative abundances greater than 1% across all groups of KEGG pathways at the second level. (**c**) The prediction of the top 40 KEGG pathways in each group at the third level. (**d**) Partial differential KEGG pathways among 4 seed-related groups (seeds 20 daf, 30 daf, 40 daf and 50 daf) at the third level, and the enrichment in different groups. (**e**) Heatmap analysis of differential pathways among the 4 seed groups. Analysis of differences was based on LEfSe (LDA = 2, *p* < 0.05). See Figure 1 for the details of flower buds; young pods; 20, 30, 40 and 50 daf seeds; mature seeds and parental seeds.

**Figure 4 plants-13-00912-f004:**
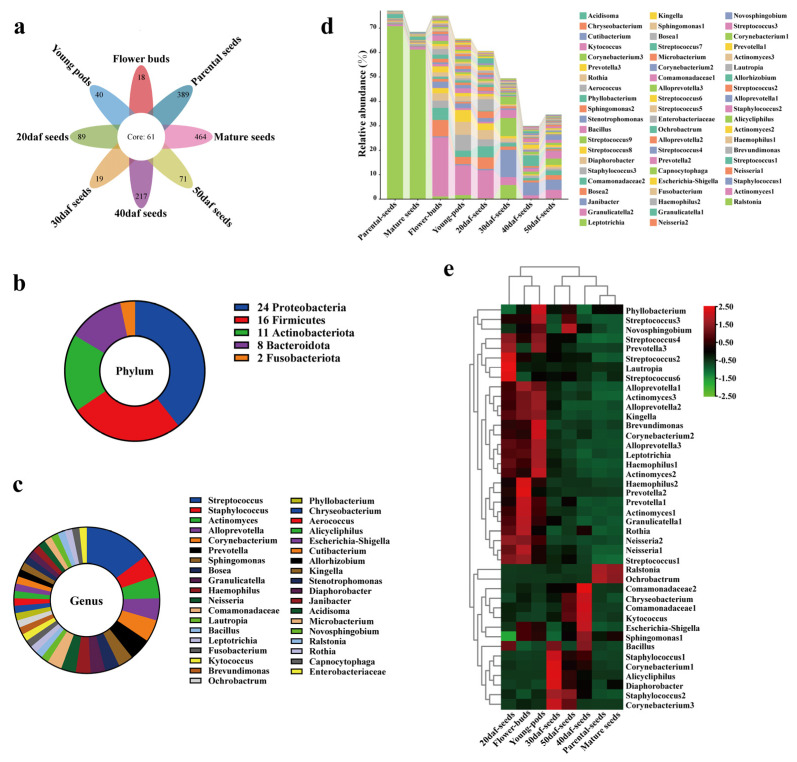
Identification of core seed microbiota from seed sowing, development to maturation. (**a**) Flower diagram showing that 61 core ASVs coexist across all groups. (**b**,**c**) The number of core microbiota at the phylum and genus levels, respectively. (**d**) Relative abundance of 61 core ASVs among all groups. (**e**) Heatmap analysis of differential core ASVs at the genus level among all groups. The analysis of differences was based on LEfSe (LDA = 2, *p* < 0.05). See Figure 1 for the details of flower buds; young pods; 20, 30, 40 and 50 daf seeds; mature seeds and parental seeds.

**Figure 5 plants-13-00912-f005:**
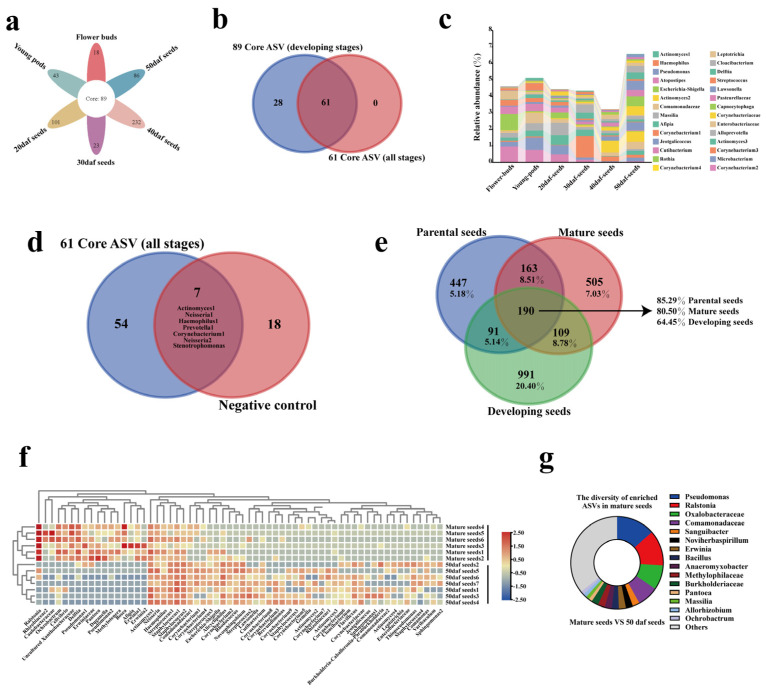
Identification of missing core microbiota, from seed sowing to maturation. (**a**) Flower diagram showing that 89 core ASVs coexisted across all groups, except for the two groups of mature seeds and parental seeds. (**b**) Venn diagram showing the ASVs missing in a comparison between the core microbiota of all stages and those of developing stages. (**c**) Relative abundance of all missing core ASVs among all groups. (**d**) Venn diagram showing the common and unique ASVs between 61 core ASVs and the ASVs of the negative control. (**e**) Venn diagram showing the number and relative abundance of common and unique ASVs among the three groups of parental seeds, mature seeds and developing seeds. (**f**) Heatmap showing the differential ASVs with a relative abundance of more than 0.01% between the groups of 50 daf seeds and mature seeds. (**g**) The number of enriched ASVs at the genus level in the group of mature seeds. Analysis of differences was based on LEfSe, with thresholds of (LDA = 2, *p* < 0.05). See Figure 1 for the details for flower buds; young pods; 20, 30, 40 and 50 daf seeds; mature seeds and parental seeds. Developing seeds comprises the six groups (flower buds and young pods, and seeds 20 daf, 30 daf, 40 daf and 50 daf) in one group. Negative control: the samples without plant tissue, but which were subjected to the same processes as seed-related samples.

**Figure 6 plants-13-00912-f006:**
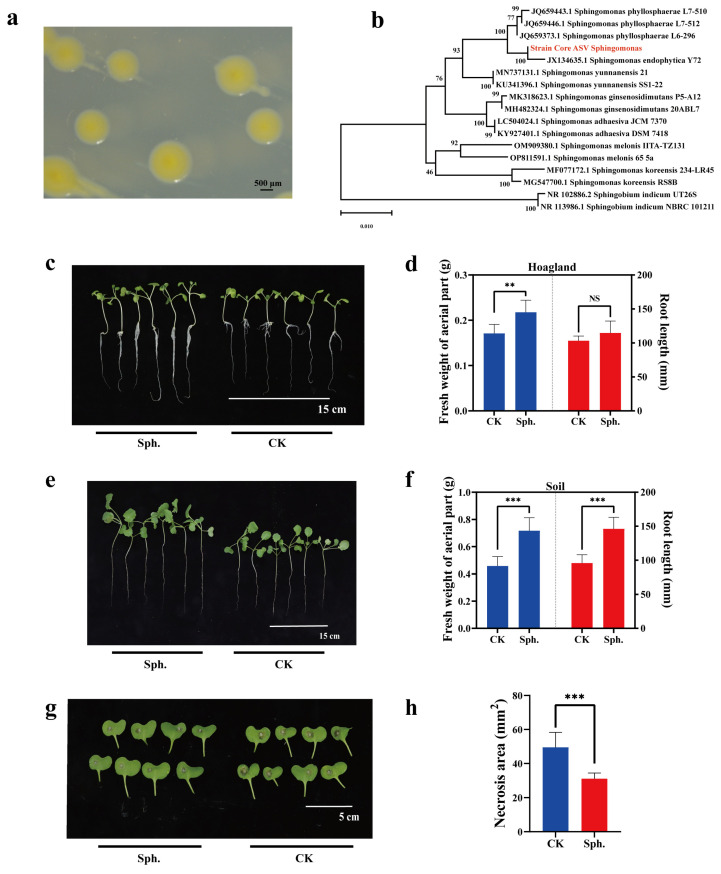
Colony morphology of *Sp. endophytica* and the effect of *Sp. endophytica* on rapeseed. (**a**) Colonies were developed on R2A medium for 5 days at 25 °C. (**b**) Evolutionary relationships of taxa: The evolutionary history was inferred using the neighbor-joining method using MEGA 11. (**c**,**e**) The morphology of rapeseed seedlings growing in Hoagland medium and soil for two weeks, and kept at 20 °C, respectively, aiming to compare the differences between the inoculated group and the control group. (**d**,**f**) Statistical data of fresh weight of aerial parts and root length of seedlings growing in Hoagland medium and soil conditions, respectively. (**g**) Lesion size of leaves after inoculation with 2-mm diameter hyphal agar disc of strain EP-1PNA367 and being kept for 24 h at 20 °C to evaluate the resistance of seedlings after seed inoculation with *Sp. endophytica*. (**h**) Statistical data from lesion areas of seedlings to *Sc. sclerotiorum*. Seedlings belonging to the *Sp. endophytica* inoculation group are on the left, and seedlings belonging to the control group are on the right. Statistical analysis of the fresh weight of aerial parts, root length and lesion areas. The data are shown as the mean ± S.D. CK represents PBS priming; Sph. represents *Sp. endophytica* bio priming. The symbol “**” indicates *p* < 0.01, the symbol “***” indicates *p* < 0.001, and the symbol “NS” indicates “no significance”. Data analysis was based on one-way ANOVA.

## Data Availability

All the raw sequencing data of 16S rRNA gene amplicons from this research are available in the NCBI Sequence Read Archive (SRA), accession number (PRJNA1007368).

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
