# Peer review of "The Dynamic Changes of Brassica napus Seed Microbiota across the Entire Seed Life in the Field"

_plants, 2024, doi:10.3390/plants13060912_

Round 1

Reviewer 1 Report

Comments and Suggestions for Authors

My congratulations to the authors, the work is very good, the experiments are adequate and answer each research question.

I just have a couple of recommendations/questions.

84: The dataset was normalized: Please specify which normalization method was used.

90: “… which was relatively lower than that of 90 seed related samples”. Please add a bibliographic reference that refers to that value

319: Correlate the species that you found with bibliographic references that indicate the resistance of those genera to environmental changes.

357: Please, include reference

368: Please include a paragraph ruling out human contamination when taking samples.

378: Explain why, despite being bacteria that are pathogenic to humans (Neisseria), bacteria in seeds maintain their abundance in different generations in flower buds, young pods, 20 daf seeds, 30 daf seeds, 40 daf seeds, 156 50 daf seeds, mature seeds, and parental seeds

410: Please give a brief explanation why only Sp. endophytica was isolated and analyzed.

Please consider exploring the possibilities of Prevotella within the seed and its role in breaking down cellulose.

In the analysis of Functions of seed microbiota based on PICRUSt2, did you find any correlation with sucrose-degrading enzymes?

Reviewer 2 Report

Comments and Suggestions for Authors

Comment#

The article ‘The Dynamic Changes of Brassica napus Seed Microbiota Crossing the Entire Seed Life History in the Field’ by Yao et al. provides information about the role of microbiota in Brassica napus seed development. They performed microbiota profiling of flower buds, pods, and seeds sampled at different stages of development using next-generation sequencing technology, downstream analyses with QIIME2, and diversity analyses with R statistical ecological packages. Metabolic profiling of the microbiota with PICRUSt2 suggests the correlation of predicted metabolite with the seed development. Moreover, they isolated Sphingomonas endophytica from seeds and found it to promote seedling growth.

The current study provides valuable information about the microbial communities associated with Brassica napus seed development and possible horizontal and vertical transmission of microbes.

The manuscript requires major English language corrections. Several sections of the manuscript are poorly written and not easy to understand.

Major Comments#

Line 19 – Change ‘branches’ to ‘groups’ (See comments below regarding this)

Line 87 – Clarify “Furthermore, we extracted kit DNA”?

Line 97-98 – The message is not clear. Rephrase it.

Lines 93- 107- What do all these alpha diversity indices signify? The reader would benefit from understanding the meaning of observed values, evenness, and the Shannon index.

Line 103 – Remove ‘tree’

Line 107—The phylogenetic terminology is incorrect here. Change “clustered into three branches” to "clustered into three groups.”.

Lines 106-110 – Use the term ‘group’ instead of ‘branches’ and rephrase the sentences accordingly

Line 107 – Figure 1e could be shown better to delimit the three groups by three colored vertical lines or name three lines with group names

Lines 162-165 – Same comment as lines 107.

Lines 184-185 – The sentence is not complete.

Lines 187-189 – I am unable to understand what the authors want to say here. Can you clarify the message from the study referenced in 65 and explain how it is relevant here?

Restructuring the entire sentence might clarify it.

Line 197 – The sentence is not complete or consider removing “For flavonoids related substances, an important class of secondary metabolites.”

Line 201 – remove ‘all’

Figure 4d is not informative. A supplementary table with the relative abundance of each ASV is sufficient.

Line 221 – Which statistical analysis was performed here?                                                                                                                 

Line 227 – Same as the comment line 221

Line 238-239 – What was the purpose of removing the mature and parental seed groups? Starting the paragraph with the objective of the analyses would benefit the readers.

Line 237 – Need better caption “2.5 Missing ASVs with seed maturation and common ASVs between two generations”

Line 273 – Change to ‘with seed development’

Lines 323- 326 - Why have the authors not discussed the possibilities of vertical and horizontal transmission of microbes? I recommend such possible scenarios here.

Line 327 – What is material accumulation?

Line 350 – Change to ‘We identified a core microbiota composed of 61 ASVs from a total of 2496 ASVs’

Line 478 – Change ‘priming sequences were cut out according to the number of bases’ to primer sequences were trimmed out according to the number of bases’

Line 481-482—Removing the sequences using the keywords' chloroplast and mitochondria' from the amplicons is new to me. Did the authors do it manually? However, QIIME2 can remove those ASVs before any downstream analyses.

Comments on the Quality of English Language

The manuscript requires major English language corrections. Several sections of the manuscript are poorly written and not easy to understand.

Round 2

Reviewer 2 Report

Comments and Suggestions for Authors

The authors improved the manuscript substantially.

Line 334-336 - Change to 'Furthermore, the seed microbes can be acquired from the environment either prior to or after the maturation of the seeds (horizontal transmission),'

Lines 390-392 – Change to ‘All plants were grown naturally in the field, and strict surface sterilization was carried out to rule out human contamination.’

Lines 533-535 – I agree with the authors that QIIME2 has the command to remove the sequences of mitochondria and chloroplast. For clarity, please consider rewriting the sentence to “Before analyses, sequences matching to ‘chloroplast’ and ‘mitochondria’ were removed by the method of taxonomy-based filtering implemented in QIIME2.”

Comments on the Quality of English Language

English editing is required
